# Phenotypic and Genomic Characterization of the *Comune di Sicilia* Goat: Towards the Conservation of an Endangered Local Breed

**DOI:** 10.3390/ani13203207

**Published:** 2023-10-13

**Authors:** Arianna Bionda, Vincenzo Lopreiato, Annalisa Amato, Matteo Cortellari, Carmelo Cavallo, Vincenzo Chiofalo, Paola Crepaldi, Luigi Liotta

**Affiliations:** 1Dipartimento di Scienze Agrarie e Ambientali—Produzione, Territorio, Agroenergia, University of Milan, Via Celoria 2, 20133 Milan, Italy; arianna.bionda@unimi.it (A.B.); paola.crepaldi@unimi.it (P.C.); 2Dipartimento di Scienze Veterinarie, University of Messina, Viale Palatucci 13, 98168 Messina, Italy; vincenzo.lopreiato@unime.it (V.L.); annalisa.amato@unime.it (A.A.); carmelo.cavallo@studenti.unime.it (C.C.); vincenzo.chiofalo@unime.it (V.C.); luigi.liotta@unime.it (L.L.)

**Keywords:** local goat breeds, goat morphology, goat genomics, anotia, wattles, breed conservation

## Abstract

**Simple Summary:**

The Comune di Sicilia goat, originating in Sicily (Italy), is in the process of being officially acknowledged as a breed. To better characterize this population, this study examined 78 goats from two locations, recording the goats’ morphological traits and measurements. Moreover, these goats were genotyped using a medium-density SNPchip and compared with goats from 15 different Italian breeds. The study found that the Comune di Sicilia goats have unique physical and genetic characteristics that distinguish them from other breeds in the same area. However, moderate variability was observed, likely influenced by how the goats were chosen by breeders. Additionally, the genomic comparison of goats with different traits led to the identification of genes possibly linked to anotia and wattle presence in goat species. To preserve this breed and avoid problems with inbreeding, a combined approach integrating phenotypic and genomic analyses is warranted.

**Abstract:**

The Comune di Sicilia, a local goat breed from Sicily (Italy), is currently undergoing recognition as a distinct breed. This study aims to characterize the population both phenotypically and genomically to advance its recognition process. A total of 78 subjects from two locations were enrolled, and their phenotypic data, including qualitative traits and morphometric measurements of adult animals, were recorded and statistically analyzed. The goats were genotyped using the Illumina 50 k Goat SNPchip, comparing them with 473 goats from 15 Italian breeds. Population structure, phylogenetic relationships, admixture, and genomic inbreeding were analyzed. Additionally, subjects with different morphological traits were compared using F_ST_ and runs of homozygosity, leading to the identification of potential candidate genes associated with anotia and wattle presence in goats. The Comune di Sicilia breed exhibited distinctive genomic and phenotypic features, setting it apart from other breeds in the same region. However, moderate variability, possibly influenced by selection practices, was also observed. To ensure the breed’s preservation and prevent excessive inbreeding, a comprehensive approach considering both morphology and genomic background is recommended. This study contributes valuable insights into the genetic peculiarities of the Comune di Sicilia goat, supporting its recognition as a unique and valuable breed.

## 1. Introduction

In recent years, the environment was influenced by globalization, urbanization, population growth, global warming, and climate change. Consequently, local breeds were affected as well, and biodiversity is under threat, making it evident that ensuring the sustainability of livestock breeding necessitates the surveillance and preservation of native breeds that are well adapted to the local environment [1,2,3], hence the emergence of the need to preserve, maintain, sustainably utilize, recover, and enhance the components of biological diversity [4]. In this context, local livestock populations play an increasingly important role, as their breeding often sustains the economies of marginal areas that would otherwise be abandoned [5], and represents an interesting alternative for the valorization of typical quality products linked to their place of origin, thus generating an ecologically sustainable livestock economy [6,7]. In particular, in Southern Italy, goat breeding is traditionally practiced with native breeds, well adapted to the environment and able to exploit and enhance it [8]. In Sicily there are about 92,714 [9] goats, including important native breeds (Argentata dell’Etna (3260, ARG), Girgentana (2603, GIR), Messinese (8814, MES), Maltese (988, MAL), and Derivata di Siria (840, DDS)) that are well adapted to marginal areas and are able to produce in the harsh conditions of this region [10,11]. However, in addition to the officially recognized breeds in possession of a registry, there is a small population (around 500 heads) called “Comune di Sicilia” (CCS), bred in the western area of Sicily (Figure 1). It was first reported by Chicoli (1870) [12], who described both its phenotype and production characteristics in his book “*Riproduzione, Allevamento e Miglioramento degli Animali Domestici in Sicilia*” (“Reproduction, Breeding and Improvement of Domestic Animals in Sicily”). Chicoli described these goats as long- and wire-haired, with a variety of possible coat colors (white, black, brown, and honey), a large head, a rather large size, and very developed udders in females, with a daily production of about three litres of milk.

Historically, studies aimed at describing livestock populations primarily relied on phenotypic and historical data. However, recent advancements in molecular tools now offer the opportunity to augment these conventional zootechnical evaluations, providing a more comprehensive and accurate characterization of animal breeds and facilitating an improved recognition process. In this context, the BIOSAVE project, “Use of phenotypic and genomic descriptors for the recovery, definition of genetic originality, and zootechnical management of Sicilian endangered local breeds”, was approved in 2021 and financed by the PSR Sicilia 2014–2020—Sub-measure 10.2b “Support for the conservation of genetic resources in agriculture and forestry”. The overall aim of this project is to ascertain the official recognition and subsequent developmental significance of Sicilian livestock breeds in relation to land, landscape, and sustainability. It also seeks to explore the role of public policies and the multifunctional approach involving research institutions in supporting these local breeds. Hence, based on the Chicoli’s first description of the “Comune di Sicilia” goat [12], and with the support of the BIOSAVE project, the aim of the present study was to define the phenotypic and genomic characteristics of the “Comune di Sicilia” goat, given its presence and historical role in the territory, as an indispensable element for a possible future opening of the appropriate genealogical register.

## 2. Materials and Methods

This study was performed according to the ethical principles that have their origins in the Italian Veterinarians’ Ethical Code [13] and the Italian and European regulations on animal welfare (Directive 2010/63/EU 2010).

### 2.1. Description of the Study Area and Animal Management

The study was conducted in western Sicily, in two farms located in Petralia Sottana and Bolognetta, two administrative areas in the province of Palermo representing different agro-ecological areas of Sicily. Petralia Sottana is part of the Madonie Park, with a warm and temperate climate, and is located at an altitude of 1039 m above sea level, specifically between latitude 37°48′0″ N and longitude 14°5′0″ E. This mountainous area is characterized by large extents for grazing with extensive arable crops. The territory of Bolognetta is mainly hilly and characterized by a warm and temperate climate; it is located between latitude 37°57′39″ N, longitude 13°27′78″ E, and altitude 348 m above sea level. In both farms, goats were reared under a semi-extensive farming system, where feeding management is based on grazing spontaneous fodder essences during the day, and during the night, shelter is provided in the stable called “mannara” (from Arabic ‘manzrah’: closed area), which in the local dialect refers to a traditional enclosure where sheep and goats are usually penned at night [14].

### 2.2. Phenotypic Data Collection and Statistical Analysis

A total of 78 goats (comprising 9 bucks and 69 does) exhibiting the most typical phenotypic characteristics of the Comune di Sicilia (CCS) goat population were carefully selected from the two herds. Specifically, 25 does and 6 bucks were selected from Petralia Sottana farm, and 44 does and 3 bucks from Bolognetta farm.

Morphometric linear measurements were taken only on the 41 adult subjects (4 bucks and 37 does with an age comprised between 18 months and 5 years); specifically, their age was estimated through the evaluation of the dentition [15,16]. Data were scored on eleven morphometric traits following the descriptor list of FAO (2012) and Abd-Allah et al. (2019) [17,18] for the phenotypic characterizations of goats.

Accordingly, the following traits were recorded using a graduated stick and a measuring tape and expressed in centimeters (cm):Wither height (WH) was measured as the vertical distance from the top of the withers to the ground.Croup height (CrH) was measured as the vertical distance from the top of the croup to the ground.Chest height (ChH) was measured as the vertical distance from sternum to withers.Chest length (ChL) was measured as the distance between the top behind the scapular and the costal arch bounded by the last rib.Trunk length (TL) measured as the distance from the point at the top behind the scapular to the base of the tail.Croup length (CrL) was measured as the distance between the iliac tuberosity and the ischial tuberosity.Chest width (CW) it was measured as the distance between the right retro-scapular area and the left retro-scapular area.Hip breadth (HB) was taken as the distance between the two iliac tuberosities.Coxofemoral width (CxW) was taken as the distance between the two trochanteric tuberosities.Hearth girth (HG) was measured as a circumferential measure taken around the chest just behind the front legs and withers.Shin circumference (SC) was measured from the left mid-metacarpus.Moreover, live body weight (BW) was calculated according to Natsir et al. (2010) [19] with the following equation: BW = 0.0127 × HG^2^ − 0.69 × HG + 14.7. According to the latter authors, heart girth measurement was determined to be the best predictor of live BW with a regression coefficient of 0.92. Measurements were recorded in the morning before the animals were released for grazing to avoid the effect of feeding and watering on the goats’ size and conformation. All measurements were performed by the same person in order to avoid inter-individual variations as reported by Sheriff et al. (2021) [20] and Arandas et al. (2017) [15]. All measurements were performed only on adult, healthy, and non-pregnant goats.

Regarding qualitative data, coat color pattern and type, presence or absence of horn, ears, wattles, and beard were also registered for all 78 subjects according to Sponenberg et al. (1998) and Henkel et al. (2019) [21,22].

Phenothypic data were analyzed with SAS software (release 9.4, SAS Institute Inc., Cary, NC, USA). The univariate procedure of SAS was used to determine the following descriptive statistics for the distribution of the morphometric values (*n* = 37 female goats) obtained: mean, first quartile, median, third quartile, standard deviation, 95% confidence interval of the mean, skewness, kurtosis, and Shapiro–Wilk test to assess if data were normally distributed.

All morphometric data were analyzed with ANOVA mixed models using the GLIMMIX procedure of SAS. The statistical models included the fixed effects of the farm (1: Bolognetta farm; 2: Petralia Sottana farm), horn (yes or no), and wattles (yes or no). Individual goats were included as random effect. In addition, for the ANOVA analysis, trunk length (TL), chest width (CW), and shin circumference (SC) were log-transformed and presented as back-transformed data due to their not-normal distribution. Pair-wise comparisons were performed using the least significant difference test. Statistical significance was declared at *p* ≤ 0.05.

### 2.3. Genomic Analyses

The genomic analyses were performed on 78 CCS goats, consisting of 9 bucks and 69 does. This group comprised all the animals that underwent morphological evaluation as previously described, along with additional subjects selected to ensure a representative sample of the breed. Moreover, special attention was given to limit direct relatedness among the individuals in the cohort. Blood samples were collected and about 3 mL of each sample was placed in a sterile tube containing ethylenediamine tetra-acetic acid (EDTA) and stored in the refrigerator or freezer until analysis. The DNA extraction and genotyping were outsourced and performed using the Goat 60 K SNP BeadChips on an iScan System (Illumina^®^, San Diego, CA, USA). Experimental protocol was authorized by the Regional Department of Agriculture, Rural Development and Mediterranean Fisheries—Sicilian Region (Dipartimento Regionale Agricoltura Assessorato Regionale dell’Agricoltura, dello Sviluppo Rurale e della Pesca Mediterranea Regione Siciliana) Italy, n. G49J21006760009, prot. 0012062, 14 July 2021.

Genomic data of CCS goats were compared to 437 goats belonging to potentially related breeds coming from data previously published by Cortellari et al. (2021) [3] (Table 1).

PLINK software (version 1.9) [23] was used to screen the genotypes and retain only individuals with a minimum call rate of 95% and SNPs located on autosomes with a minimum call rate of 95% and a minor allele frequency (MAF) of 0.1%. In addition to the previous steps, the genomic data were utilized to examine the relatedness among all subjects, and any directly related animals were excluded from the analysis. Using BITE software (version 1.1) [24], each goat population was reduced in number to a maximum of 35 subjects.

PLINK 1.9 was used to perform a multidimensional scaling analysis (MDS) to visualize the genetic distances among the goat populations included in the study. In-house scripts were used for computing bootstrapped Reynolds distances among breeds [25] and identity-by-state (IBS) distances among single individuals and creating dendrograms based on them. The genetic admixture of all individuals, representing their genetic ancestry, was analyzed using ADMIXTURE 1.3 [26], with the number of genetic clusters (K) ranging from 2 to 16. The best-fitting K was determined by the lowest cross-validation value (c-v). Individual ancestry fractions (Q-values) were also examined.

To elucidate the genetic diversity of the analyzed breeds, expected heterozygosity (He), observed heterozygosity (Ho), and Wright’s fixation index (F_IS_) were calculated using PLINK 1.9. A sliding window approach was used to estimate runs of homozygosity (ROH) in all subjects using the following parameters: ROHquartile = 0.99, minNsnp = 10, maxNsnp = 30, windef = 20, interval = 5, hetallowed = 0, minKblength = 1000, density = 500, maxInternalGap = 500, and maxmiss = 2. The ROH-based inbreeding coefficient (F_ROH_) was calculated by dividing the total length of ROH in a subject by the total length of the autosomes covered by the SNPs, as described by McQuillan et al. (2008) [27,28]. This parameter was calculated for the total ROH and for five different classes of ROH length to estimate the timing of past breeding events: 1–2 Mb, 2–4 Mb, 4–8 Mb, from 8 to 16 Mb, and >16 Mb.

The genomic effective population size (Ne) trend, ranging from 13 to 983 generations ago, was estimated using the linkage disequilibrium (LD) method for all the populations using SneP software (version 1.1) [29].

In addition, we investigated the selection signatures associated with specific morphological features found in CCS goats, such as microtia and the presence or absence of wattles and horns using F_ST_ and ROH analyses. Specifically, the following groups were compared: (a) 39 horned vs. 39 polled goats (25 and 21 from Bolognetta and 14 and 17 from Petralia Sottana farm, respectively); (b) 74 goats with ears (46 from Bolognetta and 27 from Petralia Sottana farm) vs. 4 presenting microtia (all from Petralia Sottana farm); and (c) 44 goats with wattles vs. 34 without wattles (18 and 29 from Bolognetta and 16 and 15 from Petralia Sottana farm, respectively). In particular, the SNPs falling in the top 1% F_ST_ values and delta H-score (difference in the proportion of animals in each group presenting a given ROH) were retained and mapped on the ARS 1.2. Their associated genes were further investigated.

## 3. Results and Discussion

### 3.1. Phenotypic Data

Qualitative phenotypic data were assessed in 78 goats (Table 2). Most of the population showed medium hair length (79%), while few animals had short hair (21%). The most frequent observed coat color pattern in the study area was badger face with different grades of pheomelanic dilution. Highly diluted pheomelanin (ranging from white to very light tan) was the most represented (about 49%, Figure 2A), whereas a moderate dilution (Figure 2B) was found in 29% of the subjects and undiluted pheomelanin (dark red, Figure 2C) in 22%. Moreover, the black face markings were particularly extended in some animals and covered the whole forehead, forming a mask (Figure 2C). The predominance of animals showing a diluted color may be attributed to the breed characteristics itself or the owner’s preference for light coat color as it is important for the adaptation of environment; in fact, a light coat reflects 60% of direct solar radiation in comparison to a dark color [30] with a less absorption of heat.

Wattles were present in 44% of the analyzed goats. Additionally, 40% of the individuals, both male and female, were horned, in most cases presenting spiral or lyre horns. It is noteworthy that the presence of horns in goats is advantageous for self-defense and thermoregulation [31] and seems to be associated with a better reproductive performance [32]. In fact, being hornless is associated with intersexuality and to a physiological defect known as polled intersex syndrome (PIS), which directly affects the reproduction and other phenotypic traits. However, it is interesting to note that although most of the evaluated heads (especially males) were polled, breeders reported no reproductive problems, and at a visual examination, external genitalia were normally formed. The absence of the auricular pinna (anotia) was observed in a small number of goats.

Table 3 summarizes the morphological traits of adult female CCS goats, whereas the results for male subjects are reported in Appendix A. The measurements of body weight (BW), heart girth (HG), croup height (CrH), chest height (ChH), wither height (WH), chest length (ChL), croup length (CrL), hip breadth (HB), and coxo-femoral width (CxW) were normally distributed according to Shapiro–Wilk test (*p* > 0.05). However, the trunk length (TL), chest width (CW), and shin circumference (SC) showed a not normal distribution (*p* < 0.05): TL presented a left-skewed asymmetrical and slightly leptokurtic distribution; CW was right-skewed; and SC was leptokurtic.

Comparison of quantitative and qualitative traits between the two farms are summarized in Appendix A. There were statistical differences between farms in CrH, WH, CW, CxW (farm, *p* < 0.05), and a tendency for SC (farm, *p* = 0.07), whereas there were not observed statistical differences for other morphological traits. No significant effects on the morphological measurements were instead found between animals with and without horns and with and without wattles.

When comparing the CCS goats’ average body traits with those reported in the breed standards of the other native Sicilian goats (Girgentana, GIR; Messinese, MES; and Argentata dell’Etna, ARG) [33], some differences were observed among the breeds. In fact, the GIR goats are the heaviest (on average, GIR = 46 kg, MES = 38 kg, ARG = 38 kg, and CSS = 44.7 kg), with the greatest HG (on average, GIR = 94 cm, MES = 80 cm, ARG = 80 cm, and CSS = 82.5 cm), WH (on average, GIR = 80 cm, MES = 67 cm, ARG = 67 cm, and CSS = 70.1 cm), TL (on average, GIR = 95 cm, MES = 64 cm, ARG = 66 cm, and CSS = 73.7 cm), and CW (on average, GIR = 28 cm, MES = 17 cm, ARG = 18 cm, and CSS = 20.6 cm) compared to the other Sicilian breeds; whereas, regarding the ChH, there was not observed much difference between breeds taken into account for the comparison (on average, GIR = 35 cm, MES = 31 cm, ARG = 32 cm, and CSS = 35.3 cm).

These results, in line with historical references of the breed [12], highlight unique phenotypic traits compared to other Sicilian goat breeds.

The observed morphologic variability might be attributed to lack of a systematic selection program in the breed that would help for setting up specific selection criteria. Moreover, the differences between the two sampled farms might depend on the management system, genetic by environment interaction, the breed characteristic itself, or the presence of strains within the breed. The existed variation is an opportunity for sustainable improvement, conservation, and utilization work that would be designed for this breed.

### 3.2. Genomic Population Structure and Inbreeding

Following quality control and the removal of direct relatives, a total of 487 animals and 48,039 SNPs were retained and used for ROH and selection signature investigation. To ensure homogeneity across the 15 Italian breeds, a maximum of 35 animals were considered for each breed, resulting in a final dataset of 411 subjects for population structure analyses (Table 1).

The results of multidimensional scaling (MDS) analyses are presented in Figure 3. Regarding the CCS, the majority of subjects clustered together, with only a small subset being less clearly distinguishable from other populations. Overall, the MDS plots showed distinct clustering of almost all breeds, with CAM being the most isolated one. GIR and ASP were also separated from other southern Italian populations, although ASP showed more variability and some overlap with other breeds. Additionally, the second component of the MDS plot separated MAL goats from the other breeds. Interestingly, the third component clearly separated the breeds reared in Southern Italy and the isles from those living in central Italy, which is consistent with the findings of Cortellari et al. (2021) [3].

The results of the phylogenetic tree analysis based on Reynold distances (Appendix A) are consistent with the MDS analysis, with the central Italian breeds (BIA, GCI, GAR, and MON) clustering together. The CCS is closely related to ASP and GIR, which are also reared in close proximity. In addition, the tree is based on identity-by-state (IBS), and including all individual subjects (Figure 4) clearly showed that all CCS animals clustered together and were easily distinguishable from other populations. Specifically, the CCS breed was closely related to other Sicilian breeds. Notably, the only non-distinguishable pairs were MES and ARG, and BIA and GCI.

The admixture analysis revealed that K = 11 was the best-fitting number of clusters, as determined by the model’s c-v value. Appendix A reports the c-v values and the admixture plots for all the analyzed K. However, a unique genomic signature for CCS was already apparent at K = 7 (Figure 5A). At K = 11 (Figure 5B), CCS exhibited a distinct genetic background, with a Q-score for their own cluster of 59 ± 28%. Specifically, 15 (43%) of the CCS goats had a Q-score over 67%, 13 (37%) between 33 and 67%, and only 7 (20%) under 33%. Interestingly, a difference was observed between the two sampled farms, with the 10 animals from the first farm presenting a mean Q-score of 45 ± 14% and the 25 from the second farm of 80 ± 30%. When two additional clusters were added to the admixture model (K = 13, Figure 5C), a second CCS-related genomic signature was observed. Notably, this separation was not related to the farm of origin of the goats. Instead, subjects with the highest values for the two CCS-related clusters all came from the second farm, while the most admixed ones were from the first one. The observed findings of increased admixture and greater phenotypic variability at the Petralia Sottana farm can be elucidated by the composition of the sampled animal nucleus. Indeed, this caretaker breeder deliberately acquired prime representative specimens from the surrounding geographical breeding area, aiming to form a herd characterized by minimized inbreeding. However, this obviously led to a greater variability among the animals.

Collectively, these findings provide strong evidence supporting the genomic originality of the CCS breed, marking a significant advancement in its recognition process. However, the evidence of the presence of subjects with a more admixed background highlights the importance of complementing phenotypical evaluations with genomic analyses to select the most suitable breeding animals and optimize their matings.

Genetic variability and ROH were investigated for all the included breeds (Table 4 and Figure 6). CCS breed showed a slightly lower observed heterozygosity (H_o_) than expected heterozygosity (H_e_), with a F_IS_ equal to 0.012. This is in line with what was observed for most of the other Southern Italian breeds. Genomic inbreeding (F_ROH_) ranged from 1.5% (MES) to 15% (MAL). CCS, in particular, had a mean F_ROH_ equal to 6.0%, a value near most of the other breeds. Interestingly, more than 50% of the F_ROH_ derived from ROH > 16 MB in NIC, MON, CCS, and ASP breeds, implying recent inbreeding events [34]. These results are consistent with the fact that despite CCS and most of the other studied breeds having ancient origins, they only underwent standardization and selective breeding in relatively recent years. As a consequence, while the breed’s current inbreeding value is under control, it remains crucial to monitor it periodically to prevent any excessive reduction in genomic variability.

The genomic effective population size (Ne) based on LD was computed for each of the studied breeds. In Figure 7, Ne values are reported for time intervals ranging from 13 to 50 generations ago. A consistent decreasing pattern in Ne is observed across all the populations under analysis. Among these populations, CCS goats exhibit one of the highest Ne, following behind ARG and SAR breeds, with values closely resembling those of the GCI breed. More precisely, CCS recorded an Ne of 181 at the 13-generation mark.

### 3.3. Selection Signatures

F_ST_ and ROH were used to compare CCS goats with different phenotypic characteristics. For these analyses, all 78 CCSs were included. All the results related to the selection signature analyses, including the complete name of the genes, are reported in Appendix A.

Comparing horned (*n* = 39) and polled (*n* = 39), we identified 480 SNPs within the top 1% F_ST_ values (0.12–0.25), which fell within 136 genes. With regard to ROH analysis, 197 SNPs on 184 genes were in the top 1% delta H-score. Three genes—*PCDHAC2*, *NRG2*, and *HBEGF*—were found by both analyses. As previously mentioned, polled animals in our study exhibited normally formed genitalia, and PIS was excluded through PCR analysis. However, to the best of our knowledge, none of the genes identified by our analyses were previously associated with horn development in goats or other species. It is plausible that a breed-specific mutation is responsible for polledness in this particular breed. For example, this was the case in cattle species, where several were identified as causative factors for polledness [35,36,37]. Thus, it would be essential to conduct further research to elucidate the genetic basis of this trait in goats and explore potential breed-specific genetic variations.

In the comparison of CCS goats with ears (*n* = 74) and with anotia (*n* = 4), 480 SNPs and 143 genes were identified with F_ST_ analysis (top 1% = 0.44–0.87), and 189 SNPs and 178 genes with ROH analysis. Three genes, namely *ATP12A*, *RNF17*, and *CENPJ*, were in common. Interestingly, one of the ROH regions, located on chromosome 7, was found to be a selection signature in La Mancha goats, a breed in which the anotia trait is fixed [38]. This ROH includes 16 genes, among them *HSPA9*, mutations of which are responsible for human Even-Plus syndrome [39]. Individuals affected by this syndrome often present microtia [39]. Notably, several other genes identified through F_ST_ and/or ROH analyses are associated with various syndromes that frequently include microtia or similar ear malformations among their symptoms according to the Human Phenotype Ontology database [40]: *CENPJ* and *RNF17*, both found by both the analyses, to primary autosomal recessive microcephaly 6 and Seckel syndrome; *TCOF1* to Treacher Collins syndrome; *EYA1* to brachio-oto-renal, branchiootic, and oto-facio-cervical syndromes; *SPEN* and *RERE* to 1p36 deletion syndrome and RERE-related neurodevelopmental syndrome; and *ORC6* to Meier–Gorlin syndrome 3 and ear-patella-short stature syndrome. Moreover, *GJB2* and *6* are associated to syndromic or non-syndromic deafness, whereas *SUFU*, which presented extremely high F_ST_ values, is considered to play a pivotal role in mammalian cochlear hair cell differentiation [41]. Despite being derived from an unbalanced sample, these findings represent a significant step in understanding the genetic basis of the anotia trait in goats and undoubtedly warrant further investigation.

F_ST_ analyses on subjects with (*n* = 44) and without wattles (*n* = 34) led to the identification of 480 SNPs on 137 genes in the top 1% (0.08–0.20). Instead, 203 SNPs on 197 genes were comprised in the top 1% of delta H-score in ROH analysis. *FAN1* and *TRPM1* genes were retained in both methods. Among the identified genes, *SLC9A9* and *NEDD4* genes were previously accounted as a potential candidate gene for wattle presence in goat [42]. Interestingly, our analyses identified both the *NEDD4* gene (included in ROH of animals with wattles) and its binding protein *N4BP1* (through F_ST_), further supporting their potential roles in the development of wattles in goats. Another study, instead, suggested possible association between the same phenotype and *CSMD1* [43], which emerged from our analyses as well. Additionally, two other genes might be relevant: *KIF7*, whose mutation causes acrocallosal syndrome in humans, often presenting preauricular skin tags [44], and *ADAMTSL3*, known to influence the shape of the comb in chickens [45].

## 4. Conclusions

Local livestock populations, such as the Comune di Sicilia goat breed, play a crucial role in the preservation of rural communities in harsh and marginal areas, ensuring income for these communities and contributing to the preservation of the territory.

The investigation of the Comune di Sicilia goat revealed that, presently, this population demonstrates both genomics and phenotype distinctiveness, setting it apart from other breeds reared in the same breeding ranges, representing a step forward in its recognition process. However, it is important to acknowledge that some degree of variability exists within the population, likely influenced by breeders’ selection preferences and the relatively recent initiation of a formal selection process.

To ensure the recognition and conservation of this population, a combined approach is imperative, taking into account both the morphology and appearance of the animals as well as their genomic background. This comprehensive evaluation will enable the identification and choice of the most suitable individuals, promoting the breed’s originality while preventing excessive inbreeding.

Furthermore, the Comune di Sicilia breed presents intriguing phenotypic peculiarities that warrant further investigation from a genomic perspective. Such research can provide valuable insights into the development of these traits within the goat species.

## Figures and Tables

**Figure 1 animals-13-03207-f001:**
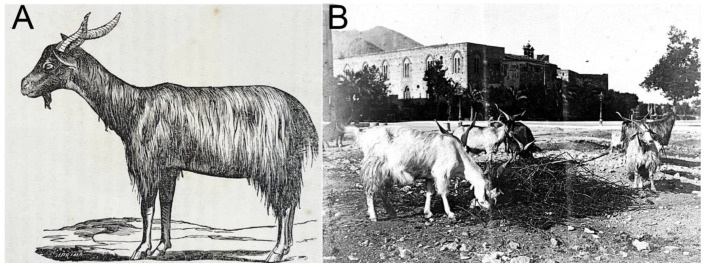
(**A**) Figure taken from “*Riproduzione, Allevamento e Miglioramento degli Animali Domestici in Sicilia*” by Chicoli (1870), representing a specimen of the so-called *Capra Comune di Sicilia*. (**B**) A historical photo of some subjects of this breed (Piazza Francesco Crispi, Palermo, 1927): it was common to see goats roaming the streets of the Sicilian cities, where a local traditional practice involved the “*capraru*” (goat breeder) delivering fresh milk at dawn by milking the goats right at people’s doorsteps.

**Figure 2 animals-13-03207-f002:**
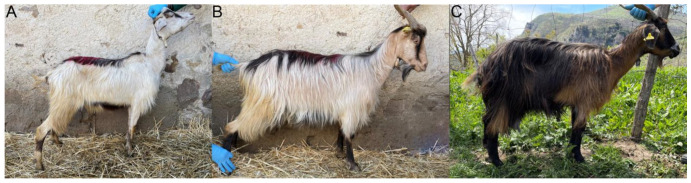
Representative coat colors observed in Comune di Sicilia breed. All the enrolled goats presented a badger face pattern, with different grades of pheomelanin dilution, from white (**A**) to tan (**B**) to dark red (**C**). In some goats, the black facial markings extended to form a mask (**C**).

**Figure 3 animals-13-03207-f003:**
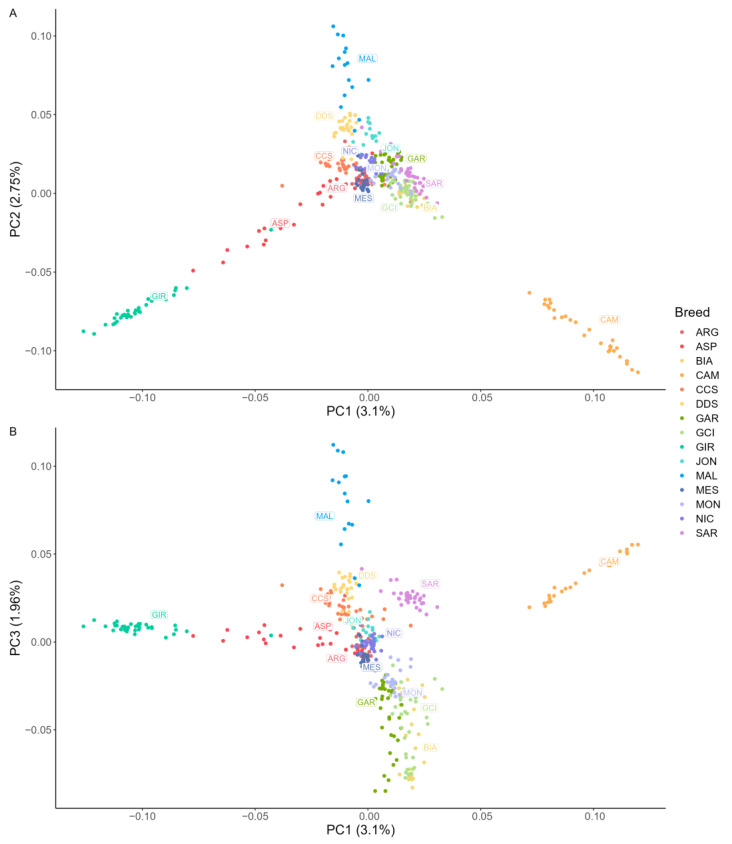
Representation of first three principal components of the multidimensional scaling analysis. Each breed is defined by a different color and each subject is represented by a dot. (**A**) Sicilian breeds: ARG—Argentata dell’Etna, CCS—Comune di Sicilia, DDS—Derivata di Siria, GIR—Girgentana, MES—Messinese. (**B**) Other Italian breeds: ASP—Aspromontana, BIA—Bianca Monticellana, CAM—Camosciata delle Alpi, GAR—Garganica, GCI—Gricia Ciociara, JON—Jonica, MAL—Maltese, MON—Capra di Montefalcone, NIC—Nicastrese, and SAR—Sarda.

**Figure 4 animals-13-03207-f004:**
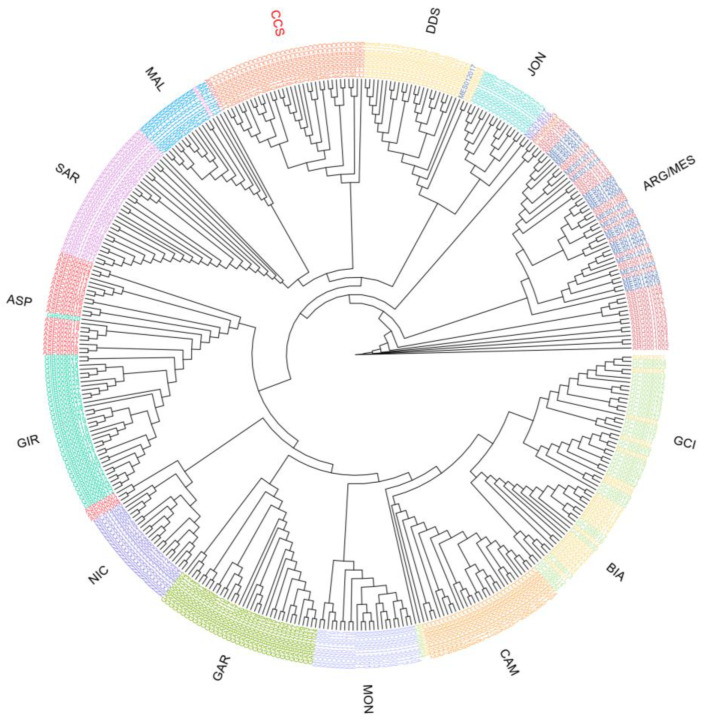
Dendrogram representing the phylogenic relationship according to identity-by-state distances. Sicilian breeds: ARG—Argentata dell’Etna, CCS—Comune di Sicilia, DDS—Derivata di Siria, GIR—Girgentana, MES—Messinese. Other Italian breeds: ASP—Aspromontana, BIA—Bianca Monticellana, CAM—Camosciata delle Alpi, GAR—Garganica, GCI—Gricia Ciociara, JON—Jonica, MAL—Maltese, MON—Capra di Montefalcone, NIC—Nicastrese, and SAR—Sarda.

**Figure 5 animals-13-03207-f005:**
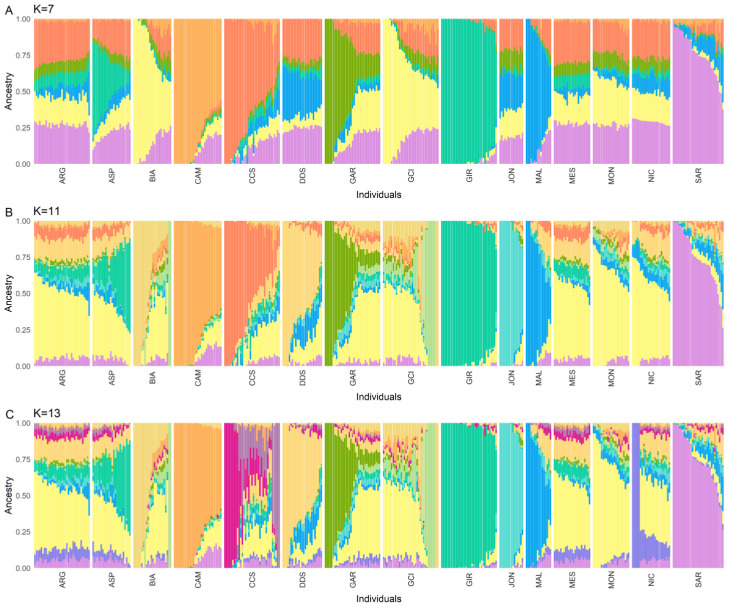
Admixture models using a number of clusters (K) equal to 7 (**A**), 11 (**B**), and 13 (**C**). The best-fitting model was found to be the one with K = 11. Each color represents a different cluster and each bar a different subject. Sicilian breeds: ARG—Argentata dell’Etna, CCS—Comune di sicilia, DDS—Derivata di Siria, GIR—Girgentana, MES—Messinese. Other Italian breeds: ASP—Aspromontana, BIA—Bianca Monticellana, CAM—Camosciata delle Alpi, GAR—Garganica, GCI—Gricia Ciociara, JON—Jonica, MAL—Maltese, MON—Capra di Montefalcone, NIC—Nicastrese, and SAR—Sarda.

**Figure 6 animals-13-03207-f006:**
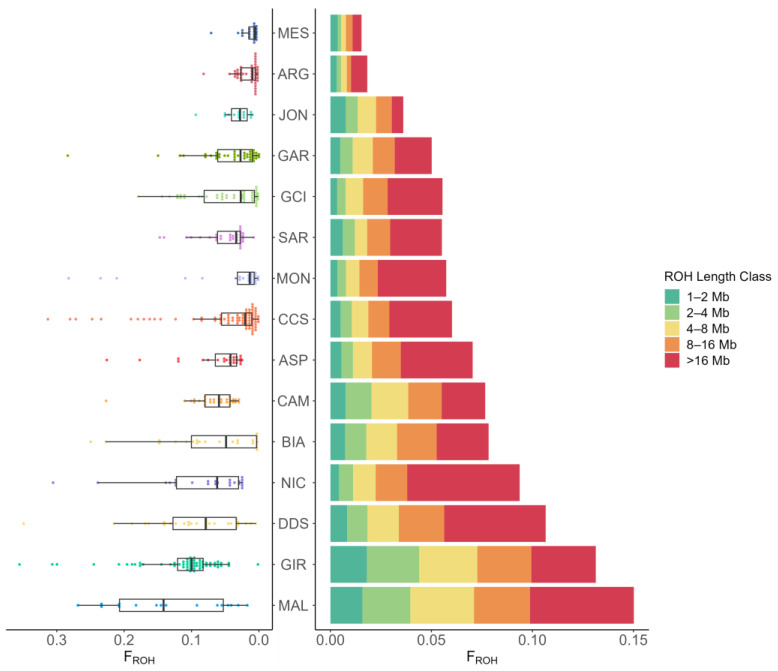
Boxplot of the total ROH-based inbreeding coefficient (F_ROH_) and barplot of F_ROH_ by ROH length by breed. Sicilian breeds: ARG—Argentata dell’Etna, CCS—Comune di sicilia, DDS—Derivata di Siria, GIR—Girgentana, MES—Messinese. Other Italian breeds: ASP—Aspromontana, BIA—Bianca Monticellana, CAM—Camosciata delle Alpi, GAR—Garganica, GCI—Gricia Ciociara, JON—Jonica, MAL—Maltese, MON—Capra di Montefalcone, NIC—Nicastrese, and SAR—Sarda.

**Figure 7 animals-13-03207-f007:**
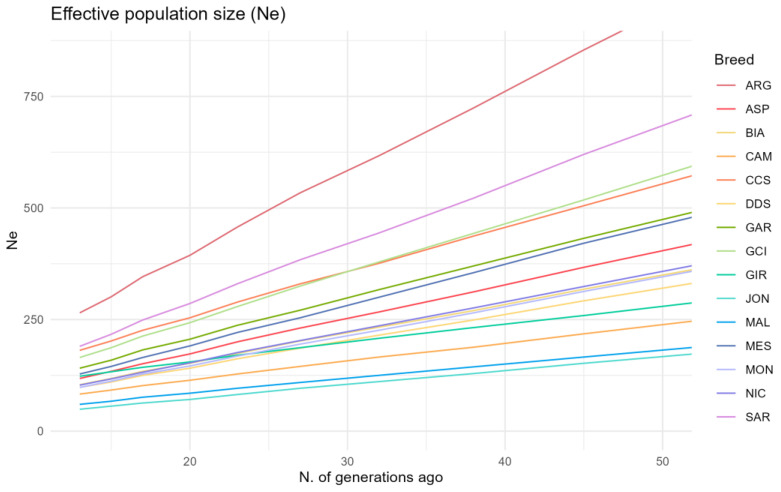
Genomic effective population size (Ne) trends. Sicilian breeds: ARG—Argentata dell’Etna, CCS—Comune di sicilia, DDS—Derivata di Siria, GIR—Girgentana, MES—Messinese. Other Italian breeds: ASP—Aspromontana, BIA—Bianca Monticellana, CAM—Camosciata delle Alpi, GAR—Garganica, GCI—Gricia Ciociara, JON—Jonica, MAL—Maltese, MON—Capra di Montefalcone, NIC—Nicastrese, and SAR—Sarda.

**Table 1 animals-13-03207-t001:** Datasets used for genomic analyses.

Breed Code	Breed Name	Region of Origin	Initial Dataset	Quality Check and Exclusion of Relatives ^a^	Breed Size Reduction ^b^
ARG	Argentata dell’Etna	Sicily	48	46	35
ASP	Aspromontana	Calabria	24	24	24
BIA	Bianca Monticellana	Lazio	24	24	24
CAM	Camosciata delle Alpi	Alpine region	30	30	30
CCS	Capra Comune di Sicilia	Sicily	78	72	35
DDS	Derivata di Siria	Sicily	32	25	25
GAR	Garganica	Apulia	40	37	35
GCI	Gricia Ciociara	Campania/Abruzzo	43	40	35
GIR	Girgentana	Sicily	59	56	35
JON	Jonica	Apulia	16	15	15
MAL	Maltese	Malta / Sicily	16	16	16
MES	Messinese	Sicily	24	23	23
MON	Capra di Montefalcone	Molise	24	23	23
NIC	Nicastrese	Calabria	24	24	24
SAR	Sarda	Sardinia	33	32	32

^a^ This dataset was used for ROH and selection signature analyses. ^b^ This dataset was used for population structure analyses.

**Table 2 animals-13-03207-t002:** Description of morphological traits of the studied cohort of Comune di Sicilia goats.

		Males (*n* = 9)	Females (*n* = 69)	Total
Horns	Present	22%	42%	40%
	Absent	78%	58%	60%
Wattles	Present	33%	45%	44%
	Absent	67%	55%	56%
Coat color	White badger face	45%	49%	49%
	Tan badger face	45%	28%	29%
	Dark red badger face	10%	23%	22%
Coat length	Short	0%	23%	21%
	Medium	100%	77%	79%
Ears	Present	78%	97%	94%
	Absent (anotia)	22%	3%	6%
Ear length	Short	29%	19%	20%
	Medium	0%	9%	8%
	Long	71%	72%	72%
Ear carriage	Erect	43%	70%	68%
	Semi-erect	43%	21%	23%
	Atonic	14%	9%	9%

**Table 3 animals-13-03207-t003:** Descriptive statistics of morphological traits in adult females of Capra Comune di Sicilia breed (*n* = 37).

		Quartiles							*p*-Value ^1^
Traits	Mean	Q1	Median	Q3	SD	95% CI	Skewness	Kurtosis	Shapiro–Wilk
Body weight (BW)	44.7	39.4	44.9	49.3	8.20	42.0–47.47	−0.03	1.18	0.62
Hearth girth (HG)	82.5	79.0	83.0	86.0	5.98	80.6–84.5	−0.59	2.18	0.19
Croup height (CrH)	68.6	67.0	68.5	70.0	3.48	67.6–69.8	0.04	−0.30	0.40
Chest height (ChH)	35.2	34.0	35.0	36.0	2.02	67.5–69.8	0.35	−0.46	0.13
Wither height (WH)	70.1	69.3	69.5	72.5	3.35	69.0–71.2	0.01	−0.82	0.32
Chest length (ChL)	39.9	38.0	40.2	42.0	3.44	38.7–41.0	−0.16	−0.09	0.85
Trunk length (TL)	73.7	71.0	75.0	77.0	4.87	72.1–75.33	−0.84	0.73	<0.05
Croup length (CrL)	26.2	25.5	26.1	27.0	1.42	25.7–26.7	0.17	−0.17	0.31
Chest width (CW)	20.6	19.5	20.5	21.5	1.91	19.9–21.2	0.82	0.23	<0.05
Hip breadth (HB)	19.0	17.5	19.0	20.5	1.68	18.5–19.5	−0.20	−0.91	0.14
Coxo-femoral width (CxW)	17.6	17.2	19.5	21.0	2.87	18.2–20.1	−0.30	−0.19	0.47
Shin circumference (SC)	8.2	8.0	8.0	9.0	0.96	7.9–8.5	−0.33	2.49	<0.05

^1^ Shapiro–Wilk test to assess if data are normally distributed. The test compares the scores in the sample to a normally distributed set of scores with the same mean and standard deviation; *p* < 0.05 indicates that variable is not normally distributed.

**Table 4 animals-13-03207-t004:** Summary of the parameters related to genomic variability and ROH-based inbreeding coefficient (F_ROH_).

Breed	H_e_	H_o_	F_IS_	Mean Number of ROH	ROH Total Length	ROH Mean Length	F_ROH_ 1–2 MB	F_ROH_ 2–4 MB	F_ROH_ 4–8 MB	F_ROH_ 8–16 MB	F_ROH_ >16 MB	F_ROH_ Total
ARG	0.412	0.413	−0.003	9.826	44.892	4.076	0.003	0.002	0.003	0.002	0.008	0.018
ASP	0.401	0.399	0.004	24.417	173.243	7.602	0.005	0.006	0.009	0.014	0.036	0.070
BIA	0.398	0.390	0.017	34.875	192.638	4.331	0.007	0.010	0.015	0.020	0.026	0.078
CAM	0.392	0.403	−0.024	37.967	188.512	4.806	0.008	0.013	0.018	0.017	0.022	0.077
CCS	0.405	0.400	0.012	22.038	148.080	4.874	0.005	0.006	0.008	0.010	0.031	0.060
DDS	0.396	0.377	0.045	39.000	262.158	5.928	0.008	0.010	0.015	0.022	0.050	0.107
GAR	0.402	0.404	−0.006	22.421	123.347	4.364	0.005	0.006	0.010	0.011	0.018	0.050
GCI	0.408	0.403	0.012	17.953	136.492	7.634	0.003	0.004	0.009	0.012	0.027	0.055
GIR	0.364	0.360	0.010	75.237	323.191	4.107	0.018	0.026	0.029	0.027	0.032	0.131
JON	0.372	0.413	−0.107	24.600	88.705	3.365	0.008	0.006	0.009	0.008	0.006	0.036
MAL	0.368	0.363	0.011	72.688	369.527	4.701	0.016	0.024	0.032	0.028	0.051	0.150
MES	0.404	0.410	−0.015	10.087	37.870	3.319	0.004	0.002	0.002	0.003	0.004	0.015
MON	0.403	0.400	0.005	17.609	141.027	4.994	0.003	0.004	0.007	0.009	0.034	0.057
NIC	0.403	0.393	0.022	26.083	230.473	8.508	0.004	0.007	0.011	0.016	0.056	0.094
SAR	0.407	0.402	0.011	23.438	135.752	6.576	0.006	0.006	0.006	0.011	0.026	0.055

## Data Availability

Data is available on request.

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
