# Peer review of "Phenotypic and Genomic Characterization of the Comune di Sicilia Goat: Towards the Conservation of an Endangered Local Breed"

_animals, 2023, doi:10.3390/ani13203207_

Round 1

Reviewer 1 Report

The manuscript entitled " Phenotypic and genomic characterization of the Comune di Sicilia goat: towards the conservation of an endangered local breed" generated datasets of 78 goats from two locations, recording goats’ morphological traits and measurement. This study contributes valuable insights into the genetic peculiarities of the Comune di Sicilia goat. The topic is interesting and the analyses are performed with sufficient detail. However, in my opinion, the manuscript needs to be entirely reworked before it can be evaluated for publication. The text needs to be rewritten to increase clarity and to improve the flow of information, particularly starting from the results section onwards.

This study mainly has the following problems:

The findings of the study are interesting but some clarity on the phenotypic data determination results are required in order to demonstrate the reliability of the selected signal test results for each group.

In the materials and methods section the authors should state more clearly the reason why they have carried out specific analyses or calculations.

Line 13-14 and line 23-24 Two repeated statements

Line 79 Place the full name and abbreviation of a noun where it first appears.

The table 1 is drawn according to the formatting requirements of the article

Fst The writing format of Fst must be consistent

line219 and 221 Fig 1C might be missing

Gene names in italics

The sentence description of the full text is confused, such as line 162 and 226

Minor editing of English language required

Author Response

The manuscript entitled " Phenotypic and genomic characterization of the Comune di Sicilia goat: towards the conservation of an endangered local breed" generated datasets of 78 goats from two locations, recording goats’ morphological traits and measurement. This study contributes valuable insights into the genetic peculiarities of the Comune di Sicilia goat. The topic is interesting and the analyses are performed with sufficient detail. However, in my opinion, the manuscript needs to be entirely reworked before it can be evaluated for publication. The text needs to be rewritten to increase clarity and to improve the flow of information, particularly starting from the results section onwards. 

This study mainly has the following problems:

The findings of the study are interesting but some clarity on the phenotypic data determination results are required in order to demonstrate the reliability of the selected signal test results for each group.

Thank you for your feedback. We have included additional details in the description of the groups being compared for the identification of selection signatures (see Materials and Methods, lines 210-220). We hope that these additions address your request.

In the materials and methods section the authors should state more clearly the reason why they have carried out specific analyses or calculations.

We included more details about the meaning of the analyses we performed in Materials and Methods section.

Line 13-14 and line 23-24 Two repeated statements

Thank you for reporting it to us, we modified the sentence at lines 13-14.

Line 79 Place the full name and abbreviation of a noun where it first appears.

We have added the abbreviation after the initial mention of the breeds. Thank you.

The table 1 is drawn according to the formatting requirements of the article

Thank you, it should be in the correct format now.

Fst The writing format of Fst must be consistent

            Thank you, the occurrence mistakenly spelled has been fixed.

line219 and 221 Fig 1C might be missing

Thank you for reporting this. Figure 1 was mistakenly cited instead of Figure 2 for the sentences about coat colours and markings.

Gene names in italics

Thank you for the suggestion, they are in italics now.

The sentence description of the full text is confused, such as line 162 and 226

            We modified the mentioned sentences, as well as other parts of the text, for better clarity.

Reviewer 2 Report

The main aim of the manuscript “Phenotypic and genomic characterization of the Comune di Sicilia goat: towards the conservation of an endangered local breed” was to evaluate the phenotypic and genomic characteristics of the “Comune di Sicilia” goat and its historical role in the territory, to obtain data to support the official recognition and subsequent developmental of this breed.

The objective includes evaluate the historical role of this population in the territory. How did the authors evaluate this objective?

Why did the authors measure the morphological traits in only 41 out of the 78 animals?

The description of the eleven morphometric traits could be deleted if they are details in the cited references.

Which is the distribution of the sampled animals in both farms?

Regarding the selection signatures, more details are needed to understand what was done. For example, in Fst analysis which pairwise comparisons were done, which criteria were used to select the genes associated with the selection signature. Please, describe in the Material and Methods section. Which is the distribution of the animals from the two farms when they were grouped?

Minor changes.

Lines 18 and 33. High or low degree of variability?

In figures 4a and b, the Italian region could be included. For example, Southern Italia, Central Italia

Author Response

The main aim of the manuscript “Phenotypic and genomic characterization of the Comune di Sicilia goat: towards the conservation of an endangered local breed” was to evaluate the phenotypic and genomic characteristics of the “Comune di Sicilia” goat and its historical role in the territory, to obtain data to support the official recognition and subsequent developmental of this breed.

The objective includes evaluate the historical role of this population in the territory. How did the authors evaluate this objective?

Thank you for your valuable suggestion. Our intention with this study was to situate it within the broader context of breed recognition, a task inherently intertwined with the historical role and development of the population. We tried to incorporate these crucial aspects into our study and paper.

Nevertheless, we acknowledge that our initial wording may have been misleading, and we have taken the necessary steps to rectify this by making amendments at lines 82-86 in the manuscript.

Why did the authors measure the morphological traits in only 41 out of the 78 animals?

Thank you for your question. As indicated in line 110-114, morphological measurements were exclusively collected from fully grown adult animals, while genomic data could also be obtained from younger animals. We modified the sentence to make it clearer.

The description of the eleven morphometric traits could be deleted if they are details in the cited references.

Thank you for your suggestion. Nevertheless, we prefer to retain this description in its current form to ensure that the reader can readily grasp how the various measurements were obtained, eliminating the need to search for them in the referenced documents and delve into the exact definitions of each measurement.

Which is the distribution of the sampled animals in both farms?

We reported the distribution of the animals in material and methods section, lines 106-109 and 210-220. Thank you for your suggestion.

Regarding the selection signatures, more details are needed to understand what was done. For example, in Fst analysis which pairwise comparisons were done, which criteria were used to select the genes associated with the selection signature. Please, describe in the Material and Methods section. Which is the distribution of the animals from the two farms when they were grouped?

Thank you for your feedback. We have included additional details in the description of the groups being compared for the identification of selection signatures (see Materials and Methods, lines 213-215). The following lines explain the selection criteria for the genes, i.e., FST analysis: we considered the genes with the highest (absolute) Fst value (top 1%); ROH analysis: we calculated a Delta H-score value for each SNP (difference in the proportion of animals which presented a ROH that included that specific SNP) and we selected those SNPs with the highest values (top 1%).

Minor changes.

Lines 18 and 33. High or low degree of variability?

            We better specified this. Thank you.

In figures 4a and b, the Italian region could be included. For example, Southern Italia, Central Italia

We specified the Sicilian breeds in the Figures’ description and the region of origin/breeding of all the included breeds in Table 1, thank you for your suggestion.

Reviewer 3 Report

The author conducted an evaluation of the genetic characteristics and diversity of the Comune di Sici lia Goat in Italy. The research work provides positive support for the protection of local goat genetic resources. However, there are significant issues overall, and it is recommended to consider  major modifications. The main issues are as follows:

1.The content of the this article is too redundant and complex, without focusing on the research focus. The author must reorganize the writing logic.

2.The writing format of the entire article is chaotic.

3. We suggest that the author supplement the K2-K15 results and images of ADMIXTURE.

4. The full text is Communication, so the length of the text is too long, especially since there are too many Figures, it is recommended to keep a small number of essential ones.

5.The phenotype determination and gene data analysis in the experimental methods are too verbose, and the author must simplify them.

6. The PCA chart is not clear, it is recommended to add labels for each group.

7. Figure 4 A is unclear.

8. The conclusion writing is too redundant, it is recommended to highlight the key points and shorten the length.

Author Response

The author conducted an evaluation of the genetic characteristics and diversity of the Comune di Sicilia Goat in Italy. The research work provides positive support for the protection of local goat genetic resources. However, there are significant issues overall, and it is recommended to consider  major modifications. The main issues are as follows:

1.The content of the this article is too redundant and complex, without focusing on the research focus. The author must reorganize the writing logic.

2.The writing format of the entire article is chaotic.

Answer to requests 1 and 2: We modified several parts of the text to improve clarity, thanks for your suggestion.

  1. We suggest that the author supplement the K2-K15 results and images of ADMIXTURE.

            Thank you for your suggestion, we included all the analysed Admixture models in Supplementary figure S2.

  1. The full text is Communication, so the length of the text is too long, especially since there are too many Figures, it is recommended to keep a small number of essential ones.

            Dear Reviewer, this paper should be categorized as an Article rather than a Short Communication, as indicated both in the related description and the header above the title. If there are any instances where we mistakenly labeled it as a Short Communication, please bring them to our attention so that we can promptly rectify the error.

5.The phenotype determination and gene data analysis in the experimental methods are too verbose, and the author must simplify them.

Thank you for your feedback. We appreciate your observation regarding the section in question. In our initial manuscript, we included the complete names of the genes for the sake of scientific rigor. However, we acknowledge that this made the paragraph less reader-friendly.

To address this issue and enhance readability, we have made revisions. Specifically, we have removed the complete gene names from the paragraph since they are already presented in the tables that detail the results of the comparison between the two groups. This modification aims to simplify the content and enhance the overall clarity of the section.

  1. The PCA chart is not clear, it is recommended to add labels for each group.

Thank you, following your suggestion we added a legend next to the plot and indicated the breed codes in the figure descriptions.

  1. Figure 4 A is unclear.

In response to the suggestions made by you and another reviewer, we have made the decision to relocate Figure 4A from the main text to the Supplementary Materials. Additionally, we have increased the size of the labels for enhanced clarity.

  1. The conclusion writing is too redundant, it is recommended to highlight the key points and shorten the length.

Thank you for your feedback. Given our choice to combine the Results and Discussion sections, we have crafted a more comprehensive conclusion section to summarize all aspects of our research. Nonetheless, we have made an effort to condense this section as per your suggestion, with the hope of addressing your request.

Reviewer 4 Report

Nicely written paper with detailed phenotypic and genotypic analysis. Here are some minor suggestions to improve the quality of this paper.

* Could you provide best K value cross-validation plot as supplementary information?

* Why the ld pruning was not done on the 50K makers before population structure analysis? Does it introduce bias with closely linked markers in the population?

* Network or pathway analysis on top Fst and ROH markers/genes can provide additional information.

*Effective population size (Ne) analysis will add great information to estimate gene pool for this breed as compared to other breeds.

Author Response

Nicely written paper with detailed phenotypic and genotypic analysis. Here are some minor suggestions to improve the quality of this paper.

* Could you provide best K value cross-validation plot as supplementary information?

            Following your suggestion, we included this plot in Supplementary Figure S2. Thank you.

* Why the ld pruning was not done on the 50K makers before population structure analysis? Does it introduce bias with closely linked markers in the population?

Thank you for your feedback. The primary reason we chose not to perform LD pruning was to avoid potentially adverse impacts on our ROH analyses. We aimed to prevent any confusion among our readers by not generating multiple datasets for individual analyses. Furthermore, it's worth noting that there are several studies in the literature that also omitted LD pruning in their analyses (for instance, refer to: Stella et al 2018, Colli et al. 2018, do Prado Paim 2018, Michailidou et al 2019, Muner et al. 2021, Mukhina et al 2021).

Nevertheless, in response to your suggestion, we re-ran the analyses, incorporating LD pruning, and found that all the results pertaining to population structure remained largely consistent with those we initially presented. Therefore, for this reason, we ultimately decided not to change the pruning parameters.

* Network or pathway analysis on top Fst and ROH markers/genes can provide additional information.

Thank you for your suggestion. In reality, we did conduct such analyses, but we chose to omit them because we found only a small number of pathways, and most of them were not clearly relevant to the traits we were investigating. Since the paper already includes several different analyses, which have made it quite lengthy, and considering that this aspect was not the primary focus of our study, we prefer not to include these findings in the paper's description. However, we are more than willing to provide you with this information upon request, and any reader interested in these analyses can perform them using the complete list of identified genes we provided in Supplementary materials.

*Effective population size (Ne) analysis will add great information to estimate gene pool for this breed as compared to other breeds.

Thank you for your valuable suggestion, which we have implemented by conducting an analysis of the genomic Ne trends for all the studied breeds. You can find this addition in the manuscript at lines 207-209 and 374-384, which also includes Figure 7.

Reviewer 5 Report

This is an interesting study on a poorly characterized goat breed form Sicilia. The question was to evaluate the originality and homogeneity of the population from other ones living in Sicila. The authors integrated morphological and genetic approaches to this aim and gave significant results.

However, I have suggestions that could improve the clarity of the message. 

1. When the morphological data are presented to provide comparisons with the three major breeds living in Sicilia, no statistics were included. I suggest that you could classify the most discriminant characters between these breeds, and test if there were significant differences. 

2.  The originality of a breed is determined by morphometric and genetic measures by comparison to other breeds. But the present day state is the result of the history of the events that shaped the actual breed. More clearly, the project is to shed light on the affinities of the breed of interest. The broad question is to study whether the breed is more related to sympatric breeds than to continental or Sardinian breeds. The ROH approach should give information on ancient or recent genetic exchanges between CCS and the other breeds, as Reynolds distance is non pertinent for that. By separating the analyses based on short and long segments of ROH, you could (maybe) provide more accurate conclusions. I mean that your comments on the proximity explained from lines 298 to 310 do not convince me (they are not wrong, but I feel that they do not respond to the question on the origin of CCS). 

About the reduction off ears, please be prudent in the conclusion about genetic signature in reason of a very unbalanced sampling. 

Minor points:

Maybe it would be improve the comprehension by marking Sicilian breeds from the others in the different figures presented. 

In figure 4A, please write larger character size.

In table 4, please extend the size of the 2nd column width. 

I m not a native speaking English and it is difficult to have an pertinent advice. However, the text is easy to read. 

Author Response

This is an interesting study on a poorly characterized goat breed form Sicilia. The question was to evaluate the originality and homogeneity of the population from other ones living in Sicila. The authors integrated morphological and genetic approaches to this aim and gave significant results.

However, I have suggestions that could improve the clarity of the message. 

  1. When the morphological data are presented to provide comparisons with the three major breeds living in Sicilia, no statistics were included. I suggest that you could classify the most discriminant characters between these breeds, and test if there were significant differences. 

We are grateful and agree with your suggestion. However, as specified in the text (please, see lines 268-270, which we have modified to clarify this point) data about other breeds derived from their standards, as data about these populations were not included in the present study. Therefore, they were mentioned with the purpose of giving an idea about the overall differences among Sicilian breeds rather than making a statistical comparison, which unfortunately is not feasible with data at our disposal at the moment. However, we appreciate the idea, which would be interesting to develop in another study.

  1. The originality of a breed is determined by morphometric and genetic measures by comparison to other breeds. But the present day state is the result of the history of the events that shaped the actual breed. More clearly, the project is to shed light on the affinities of the breed of interest. The broad question is to study whether the breed is more related to sympatric breeds than to continental or Sardinian breeds. The ROH approach should give information on ancient or recent genetic exchanges between CCS and the other breeds, as Reynolds distance is non pertinent for that. By separating the analyses based on short and long segments of ROH, you could (maybe) provide more accurate conclusions. I mean that your comments on the proximity explained from lines 298 to 310 do not convince me (they are not wrong, but I feel that they do not respond to the question on the origin of CCS). 

We appreciate your intriguing suggestions. However, it is important to note that the primary objective of the current study was not to delve into the phylogenetic origins of the Comune di Sicilia goat. Instead, our focus was on providing a comprehensive description of this population from both morphological and genomic perspectives. This included an analysis of their genomic makeup and potential connections with other similar breeds found in the same breeding areas.
To clarify, our goal was to gather evidence that supports the formal recognition of Comune di Sicilia goats as a distinct breed. This recognition is more pertinent to recent admixture rather than ancient origins. Nevertheless, we acknowledge the interest in studying the breed's origins, and we believe such a study could be valuable. In such an investigation, it would be advantageous to include all other Italian breeds for a more precise comparison.

  1. About the reduction off ears, please be prudent in the conclusion about genetic signature in reason of a very unbalanced sampling. 

Following your suggestion, we specified that the results were obtained from an unbalanced sample when discussing them. We agree that further investigation is needed, but we considered intriguing that, despite the small number of goats presenting anotia -a trait that, actually was not expected to be found in this population-, many signals were in common with American LaMancha goat, despite it being a breed originated far away.

Minor points:

Maybe it would be improve the comprehension by marking Sicilian breeds from the others in the different figures presented. 

            Thank you, we included this information in the description of the figures.

In figure 4A, please write larger character size.

            We relocated figure 4A in Supplementary materials and increased the size of the labels. Thank you for your suggestion.

In table 4, please extend the size of the 2nd column width. 

            We followed your request, thank you for bringing it to our attention.